# High-Frequency-aware Hierarchical Contrastive Selective Coding for Representation Learning on Text-attributed Graphs

## ABSTRACT

We investigate node representation learning on text-attributed graphs (TAGs), where nodes are associated with text information. Although recent studies on graph neural networks (GNNs) and pretrained language models (PLMs) have exhibited their power in encoding network and text signals, respectively, less attention has been paid to delicately coupling these two types of models on TAGs. Specifically, existing GNNs rarely model text in each node in a contextualized way; existing PLMs can hardly be applied to characterize graph structures due to their sequence architecture. To address these challenges, we propose HASH-CODE, a **H**igh-frequency **A**ware **S**pectral **H**ierarchical **Co**ntrastive Selective C**d**ing method that integrates GNNs and PLMs into a unified model. Different from previous "cascaded architectures" that directly add GNN layers upon a PLM, our HASH-CODE relies on five self-supervised optimization objectives to facilitate thorough mutual enhancement between network and text signals in diverse granularities. Moreover, we show that existing contrastive objective learns the low-frequency component of the augmentation graph and propose a high-frequency component (HFC)-aware contrastive learning objective that makes the learned embeddings more distinctive. Extensive experiments on six real-world benchmarks substantiate the efficacy of our proposed approach. In addition, theoretical analysis and item embedding visualization provide insights into our model interoperability.

## KEYWORDS

Text Attributed Graph, Graph Neural Networks, Transformer, Contrastive Learning

## 1 INTRODUCTION

Graphs are pervasive in the real world, and it is common for nodes within these graphs to be enriched with textual attributes, thereby giving rise to text-attributed graphs (TAGs) [69]. For instance, academic graphs [41] incorporate papers replete with their titles and abstracts, whereas social media networks [66] encompass tweets accompanied by their textual content. Consequently, the pursuit of learning within the realm of TAGs has assumed significant prominence as a research topic spanning various domains, *e.g.,* network analysis [52], recommender systems [64], and anomaly detection [33].

In essence, graph topology and node attributes comprise two integral components of TAGs. Consequently, the crux of representation learning on TAGs lies in the amalgamation of graph topology and node attributes. Previous works mainly adopt a cascaded architecture [23, 29, 65, 71] (Figure 1(a)), which entails encoding the textual attributes of each node with Pre-trained Language Models (PLMs), subsequently utilizing the PLM embeddings as features to train a Graph Neural Network (GNN) for message propagation [8, 12, 61]. However, as the modeling of node attributes and graph topology are

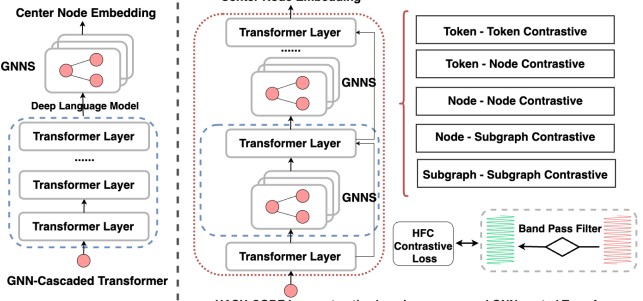

**Figure 1: (a) An illustration of GNN-cascaded transformer. (b) An illustration of our proposed contrastive learning-empowered GNN-nested transformer. The red and green twines denote the original graph signals and the mixed LFC and HFC signals from the spectral perspective.**

segregated, this learning paradigm harbors conspicuous limitations. Firstly, the link connecting two nodes is not utilized when generating their text representations. In fact, linked nodes can benefit each other regarding text semantics understanding. For example, given a paper on "LDA" and its citation nodes which are related to topic modeling, the "LDA" can be more likely interpreted as "Latent Dirichlet Allocation" rather than "Linear Discriminant Analysis". In addition, this paradigm may yield textual embeddings that are not pertinent to downstream tasks, thereby impeding the model's ability to learn node representations suitable for such tasks. Moreover, given that the formation of the graph's topological structure is intrinsically driven by the node attribute [69], this paradigm may adversely affect the comprehension of the graph topology.

Fortunately, recent efforts have been undertaken [1, 22, 37, 60] to co-train GNNs and LMs within a unified learning framework. For example, GraphFormers [60] introduces GNN-nested transformers, facilitating the joint encoding of text and node features. Heterformer [22] alternately stacks the graph aggregation module and a transformer-based text encoding module into a cohesive model to capture network heterogeneity. Despite the demonstrated efficacy of existing methods, they are encumbered by two primary drawbacks that may undermine the quality of representation learning. Firstly, these methods typically employ supervised training, necessitating a substantial volume of labeled data. However, in numerous scientific domains, labeled data are scarce and expensive to obtain [19, 53]. Secondly, these methods rely exclusively on limited optimization objectives to learn the entire model. When GNNs and LMs are jointly trained, the associated parameters are also learned through the constrained optimization objectives. It has been observed that such an optimization approach fails to capture the fine-grained correlations between textual features and graphic patterns [60, 70]. Consequently, the importance of learning graph

representations in an unsupervised or self-supervised manner is becoming increasingly paramount.

In order to tackle the aforementioned challenges, we draw inspiration from the concept of contrastive learning to enhance representation learning on TAGs. Contrastive learning [5, 15, 17, 45] refines representations by drawing positive pairs closer while maintaining a distance between negative pairs. As data sparsity and limited supervision signals constitute the two primary learning obstacles associated with existing co-training methods, contrastive learning appears to offer a promising solution to both issues: it capitalizes on intrinsic data correlations to devise auxiliary training objectives and bolsters data representations with an abundance of self-supervised signals.

In practice, representation learning on TAGs with contrastive learning is non-trivial, primarily encountering the following three challenges: (1) *How to devise a learning framework that capitalizes on the distinctive data properties of TAGs?* The contextual information within TAGs is manifested in a multitude of forms or varying intrinsic characteristics, such as tokens, nodes, or sub-graphs, which inherently exhibit complex hierarchical structures. Moreover, these hierarchies are interdependent and exert influence upon one another. How to capitalize these unique properties of TAGs remains an open question. (2) *How to design effective contrastive tasks?* To obtain an effective node embedding that fully encapsulates the semantics, relying solely on the hierarchical topological views of TAGs remains insufficient. Within TAGs, graph topological views and textual semantic views possess the capacity to mutually reinforce one another, indicating the importance of exploring the cross-view contrastive mechanism. Moreover, the hierarchies in TAGs can offer valuable guidance in selecting positive pairs with analogous semantics and negative pairs with divergent semantics, an aspect that has received limited attention in existing research [25, 57]. (3) *How to learn distinctive representations?* In developing the contrastive learning framework, we draw inspiration from the recently proposed spectral contrastive learning method [14], which outperforms several contrastive baselines with solid theoretical guarantees. However, we demonstrate that, from a spectral perspective, the spectral contrastive loss primarily learns the low-frequency component (LFC) of the graph, significantly attenuating the effects of high-frequency components (HFC). Recent studies suggest that the LFC does not necessarily encompass the most vital information [2, 7], and would ultimately contribute to the over-smoothing problem [3, 4, 31, 32], causing node representations to converge to similar values and impeding their differentiation. Consequently, more explorations are needed to determine how to incorporate the HFC to learn more discriminative embeddings.

To this end, we present a novel **H**igh-frequency **A**ware **S**pectral **H**ierarchical **Co**ntrastive Selective C**d**ing framework (**HASH-CODE**) to enhance TAG representation learning. Building upon a GNN and Transformer architecturee [60, 71], we propose to jointly train the GNN and Transformer with self-supervised signals (Figure 1(b) depicts this architecture). The primary innovation lies in the contrastive joint-training stage. Specifically, we devise five self-supervised optimization objectives to capture hierarchical intrinsic data correlations within TAGs. These optimization objectives are developed within a unified framework of contrastive learning. Moreover, we propose a loss that can be succinctly expressed as a contrastive

learning objective, accompanied by robust theoretical guarantees. Minimizing this objective results in more distinctive embeddings that strike a balance between LFC and HFC. Consequently, the proposed method is capable of characterizing correlations across varying levels of granularity or between different forms in a general manner.

Our main contributions are summarized as follows:
- We propose five self-supervised optimization objectives to maximize the mutual information of context information in different forms or granularities.
- We systematically examine the fundamental limitations of spectral contrastive loss from the perspective of spectral domain. We prove that it learns the LFC and propose an HFC-aware contrastive learning objective that makes the learned embeddings more discriminative.
- Extensive experiments conducted on three million-scale text-attributed graph datasets demonstrate the effectiveness of our proposed approach.

## 2 RELATED WORK

### 2.1 Representation Learning on TAGs

Representation learning on TAGs constitutes a significant research area across multiple domains, including natural language processing [47, 49], information retrieval [50, 58], and graph learning [59, 62]. In order to attain high-quality representations for TAGs, it is imperative to concurrently harness techniques from both natural language understanding and graph representation. The recent advancements in pretrained language models (PLM) and graph neural networks (GNN) have catalyzed the progression of pertinent methodologies.

**Seperated Training.** A number of recent efforts strive to amalgamate GNNs and LMs, thereby capitalizing on the strengths inherent in both models. The majority of prior investigations on TAGs employ a "cascaded architecture" [23, 29, 65, 71], in which the text information of each node is initially encoded through transformers, followed by the aggregation of node representations via GNNs. Nevertheless, these PLM embeddings remain non-trainable during the GNN training phase. Consequently, the model performance is adversely impacted by the semantic modeling process, which bears no relevance to the task and topology at hand.

**Co-training.** In an attempt to surmount these challenges, concerted efforts have been directed towards the co-training of GNNs and PLMs within a unified learning framework. GraphFormers [60] presents GNN-nested transformers, facilitating the concurrent encoding of text and node features. Heterformer [22] alternates between stacking the graph aggregation module and a transformer-based text encoding module within a unified model, thereby capturing network heterogeneity. However, these approaches solely depend on a single optimization objective for learning the entire model, which considerably constrains their capacity to discern the fine-grained correlations between textual and graphical patterns.

### 2.2 Contrastive Learning

**Empirical Works on Contrastive learning.** Contrastive methods [5, 6, 17] derive representations from disparate views or augmentations of inputs and minimize the InfoNCE loss [36], wherein

two views of identical data are drawn together, while views from distinct data are repelled. The acquired representation can be utilized to address a wide array of downstream tasks with exceptional performance. In the context of node representation learning on graphs, DGI [45] constructs local patches and global summaries as positive pairs. GMI [38] is designed to establish a contrast between the central node and its local patch, derived from both node features and topological structure. MVGRL [15] employs contrast across views and explores composition between varying views.

**Theoretical works on Contrastive Learning.** The exceptional performance exhibited by contrastive learning has spurred a series of theoretical investigations into the contrastive loss. The majority of these studies treat the model class as a black box, with notable exceptions being the work of [28], which scrutinizes the learned representation with linear models, and the research conducted by [42] and [54], which examine the training dynamics of contrastive learning for linear and 2-layer ReLU networks. Most relevant to our research is the study by [39], which adopts a spectral graph perspective to analyze contrastive learning methods and introduces the spectral contrastive loss. We ascertain that the spectral contrastive loss solely learns the LFC of the graph.

Different from the existing works, our research represents the first attempt to contemplate the correlations inherent within the contextual information as self-supervised signals in TAGs. We endeavor to maximize the mutual information among the views of the token, node, and subgraph, which encompass varying levels of granularity within the contextual information. Our HFC-aware loss facilitates the learning of more discriminative data representations, thereby enhancing the performance of downstream tasks.

## 3 PRELIMINARIES

In this section, we first give the definition of the text-attributed graphs (TAGs) and formulate the node representation learning problem on TAGs. Then, we introduce our proposed HFC-aware spectral contrastive loss.

### 3.1 Definition (Text-attributed Graphs)

A text-attributed graph is defined as $\mathcal{G} = (\mathcal{V}, \mathcal{E})$, where $\mathcal{V} = \{v_1, ..., v_N\}$ and $\mathcal{E}$ denote the set of nodes and edges, respectively. Let $A \in \mathbb{R}^{N \times N}$ be the adjacency matrix of the graph such that $A_{i,j} = 1$ if $v_j \in \mathcal{N}(v_i)$, otherwise $A_{i,j} = 0$. Here $\mathcal{N}(.)$ denotes the one-hop neighbor set of a node. Besides, each node $v_i$ is associated with text information.

### 3.2 Problem Statement

Given a textual attibuted graph $\mathcal{G} = (\mathcal{V}, \mathcal{E})$, the task is to build a model $f_\theta : \mathcal{V} \rightarrow \mathbb{R}^K$ with parameters $\theta$ to learn the node embedding matrix $F \in \mathbb{R}^{N \times K}$, taking network structures and text semantics into consideration, where $K$ denotes the number of feature channels. The learned embedding matrix $F$ can be further utilized in downstream tasks, *e.g.,* link prediction, node classification, *etc.*

### 3.3 HFC-aware Spectral Contrastive Loss

An important technique in our approach is the high-frequency aware spectral contrastive loss. It is developed based on the analysis of the conventional spectral contrastive loss [14]. Given a node $v$,

the conventional spectral contrastive loss is defined as:

$$\mathcal{L}_{Spectral}(v, v^+, v^-) = -2 \cdot \mathbb{E}_{v,v^+}[f_\theta(v)^T f_\theta(v^+)] + \mathbb{E}_{v,v^-}[(f_\theta(v)^T f_\theta(v^-))^2], \tag{1}$$

where $(v, v^+)$ is a pair of positive views of node $v$, $(v, v^-)$ is a pair of negative views, and $f_\theta$ is a parameterized function from the node to $\mathbb{R}^K$. Minimizing $\mathcal{L}_{Spectral}$ is equivalent to spectral clustering on the population view graph [14], where the top smallest eigenvectors of the Laplacian matrix are preserved as the columns of the final embedding matrix $F$.

In Appendix A.1, we demonstrate that, from a spectral perspective, $\mathcal{L}_{Spectral}$ primarily learns the low-frequency component (LFC) of the graph, significantly attenuating the effects of high-frequency components (HFC). Recent studies suggest that the LFC does not necessarily encompass the most vital information [2, 7], and would ultimately contribute to the over-smoothing problem [3, 4, 31, 32].

As an alternative of such low-pass filter, to introduce HFC, we propose our HFC-aware spectral contrastive loss as follows:

$$\mathcal{L}_{HFC}(v, v^+, v^-) = -2\alpha \cdot \mathbb{E}_{v,v^+}[f_\theta(v)^T f_\theta(v^+)] + \mathbb{E}_{v,v^-}[(f_\theta(v)^T f_\theta(v^-))^2], \tag{2}$$

where $\alpha$ is used to control the rate of HFC within the graph.

Upon initial examination, one might observe that our $\mathcal{L}_{HFC}$ formulation closely aligns with $\mathcal{L}_{Spectral}$. Remarkably, the primary distinction lies in the introduction of the parameter $\alpha$. However, this is not a mere trivial addition; it emerges from intricate mathematical deliberation and is surprisingly consistent with $\mathcal{L}_{Spectral}$ that offers a nuanced control of the HFC rate within the graph. Minimizing our $\mathcal{L}_{HFC}$ results in more distinctive embeddings that strike a balance between LFC and HFC. Please kindly refer to Appendix A.1 for detailed discussions and proof.

## 4 METHODOLOGY

### 4.1 Overview

Existing studies [23, 29, 65, 71] mainly emphasize the effect of sequential and graphic characteristics using the supervised optimization objective alone. Inspired by recent progress with contrastive learning [5, 17], we take a different perspective to characterize the data correlations by contrasting different views of the raw data.

The basic idea of our approach is to incorporate several elaborately designed self-supervised learning objectives for enhancing the original GNN and PLM. To develop such objectives, we leverage effective correlation signals reflected in the intrinsic characteristics of the input. As shown in Figure 2, for our task, we consider the information in different levels of granularity, including token, node and sub-graph, which are considered as different views of the input. By capturing the multi-view correlation, we unify these self-supervised learning objectives with the typical joint learning training scheme in language modeling and graph mining [60].

### 4.2 Hierarchical Contrastive Learning with TAGs

TAGs naturally possess 3 levels in the hierarchy: token-level, node-level and subgraph-level. Based on the above GNN and PLM model,

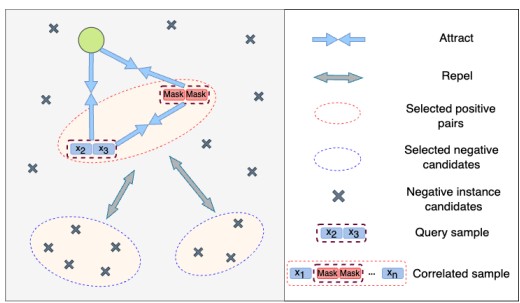

Figure 2: The overall architecture of HASH-CODE. With GraphFormers as our base model, we incorporate five self-supervised learning objectives based on the HFC-aware contrastive loss to capture the text-graph correlations in different granularities. Spectral contrastive loss learns the LFC while our HFC-aware loss achieves the balance between HFC and LFC.

we further incorporate additional self-supervised signals with contrastive learning to enhance the representations of input data. We adopt a joint-training way to construct different loss functions based on the multi-view correlation.

### 4.2.1 Intra-hierarchy contrastive learning.

**Modeling Token-level Correlations.** We first begin with modeling the bidirectional information in the token sequence. Inspired by the masked language model like BERT [10], we propose to use the contrastive learning framework to design a task that maximizes the mutual information between the masked sequence representation and its contextual representation vector. Specifically, for a node $v$, given its textual attribute sequence $x_v = \{x_{v,1}, x_{v,2}, ..., x_{v,T}\}$, we consider $x_{v,i:j}$ and $\hat{x}_{v,i:j}$ as a positive pair, where $x_{v,i:j}$ is an $n$-grams spanning from i to j and $\hat{x}_{v,i:j}$ is the corresponding sequence masked at position i to j. We may omit the subscript $v$ for notation simplification when it is not important to differentiate the affiliation between node and textual sequence.

For a specific query $n$-gram $x_{i:j}$, instead of contrasting it indiscriminately with all negative candidates $\mathcal{N}$ in a batch [27], we select truly negative samples for contrasting based on the supervision signals provided by the hierarchical structure in TAGs, as shown in Figure 3. Intuitively, we would like to eliminate those candidates sharing highly similar semantics with the query, while keeping the ones that are less semantically relevant to the query. To achieve this goal, we first define a similarity measure between an $n$-gram and a node. Inspired by [30], for a node $v$, we define the semantic similarity between $n$-gram's hidden state $h_{x_{i:j}}$ and this node's hidden state $h_v$ using a node-specific dot product:

$$s(h_{x_{i:j}}, h_v) = \frac{h_{x_{i:j}} \cdot h_v}{\tau_{h_v}}, \tau_{h_v} = \frac{\Sigma_{h_{x_i} \in H_v}||h_{x_i} - h_v||_2}{|H_v|log(|H_v| + \epsilon)},$$

where $h_{x_i}$ is the hidden representation of the token $x_i$, $H_v$ consists of the hidden representations of the tokens assigned to node $v$, and $\epsilon$ is a smooth parameter balancing the scale of temperature $\tau_{h_v}$ among different nodes.

On such a basis, we conduct negative sampling selection considering both the token and node hierarchies. Given the query $n$-gram $x_{i:j}$, we denote its corresponding node $v$'s representation as $h_v$. For a negative candidate, we are more likely to select it if its similarity

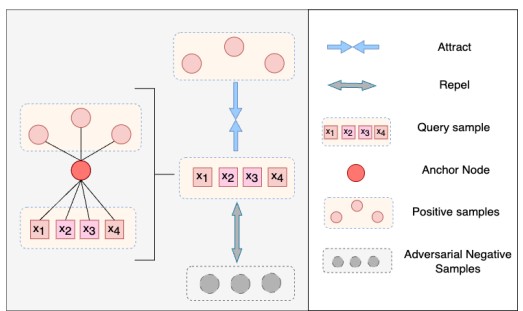

Figure 3: Token-level contrastive selective coding.

with $h_v$ is less prominent compared with other negative candidates' similarities with $h_v$. Based on such an intuition, the least dissimilar negative samples $\mathcal{N}_{select}(h_{x_{i:j}})$ are produced for the specific query.

By using these refined negative samples, we define the objective function of token-level contrastive (TC) loss as below:

$$\mathcal{L}_{TC} = \frac{1}{M}\Sigma_{m=1}^{M}\mathcal{L}_{HFC}(x_{m,i:j}, \hat{x}_{m,i:j}, \mathcal{N}_{select}(h_{x_{m,i:j}})), \quad (3)$$

where $M$ is the size of the representation set and $\mathcal{L}_{HFC}$ is our proposed HFC-aware spectral contrastive loss.

Figure 4: Modeling node-level correlations.

**Modeling Node-level Correlations.** Investigating the cross-view contrastive mechanism is especially important for node representation learning [53]. As mentioned before, nodes in TAGs possess

textual attributes that can indicate semantic relationships in the network and serve as complementary to structural patterns. As shown in Figure 4, given a node $v$, we treat its textual attribute sequence $x_v$ and its direct connected neighbors $u$, for $u \in N_v$ as two different views.

The negative selective encoding strategy used in token-level correlation modeling may select those easy negative samples that contribute less and less during the training process. Inspired by [56], we propose to adversarially generate the negative samples $\widetilde{v}$ in the node-level contrastive learning process. Specifically, we adopt ProGCL [56] method to reweight the negative node samples and performing mixup operation [67] to generate hard negative samples $\widetilde{v}$. Therefore, we minimize the following Node-level Contrastive (NC) loss:

$$\mathcal{L}_{NC} = \frac{1}{M}\Sigma_{m=1}^M \mathcal{L}_{HFC}(x_{m,v}, N_{m,v}, \widetilde{v_m}) \qquad (4)$$

**Modeling Subgraph-level Correlations.** Having modeled correlations between a node's local neighborhood and its textual features, we further consider modeling the correlations between subgraphs to cover both of the local and high-order structures of the nodes. Intuitively, nodes and their regional neighborhoods are more correlated while long-distance nodes hardly influence them. Therefore, local communities may form with the graph. This assumption is more reasonable as the size of graphs increases. Therefore, we sample a series of subgraphs including regional neighborhoods from the original graph as training data.

The most critical issue now is to sample a context subgraph, which can provide sufficient structure information for learning a high-quality representation for the central node. Here we follow the subgraph sampling based on personalized PageRank algorithm (PPR) [20] as introduced in [21, 68]. Considering the importance of different neighbors varies, for a specific node $i$, the subgraph sampler $S$ first measures the importance scores of other neighbor nodes by PPR. Given the relational information between all nodes in the form of an adjacency matrix, $A \in \mathbb{R}^{N \times N}$, the importance score matrix $S$ can be denoted as

$$S = \alpha \cdot (I - (1 - \alpha) \cdot \overline{A}),$$

where $I$ is the identity matrix and $\alpha \in [0, 1]$ is a parameter that is always set as 0.15. $\overline{A} = AD^{-1}$ denotes the colum-normalized adjacency matrix, where $D$ denotes the corresponding diagonal matrix with $D(i, i) = \Sigma_j A(i, j)$ on its diagonal. $S(i, :)$ is the importance scores vector for node $i$, indicating its correlation with other nodes.

It is noted that the importance score matrix S can be precomputed before model training starts. And we implement node-wise PPR to calculate importance scores to reduce computation memory, which makes our method more suitable to work on large-scale graphs.

For a specific node $i$, the subgraph sampler $S$ chooses top-k important neighbors to constitute the subgraph $G_i$. The index of chosen nodes can be denoted as

$$idx = top\_rank(S(i, :), k),$$

where $top\_rank$ is the function that returns the indices of the top k values and k denotes the size of context graphs.

The subgraph sampler $S$ will process the original graph with the node index to obtain the context subgraph $G_i$ of node $i$. Its

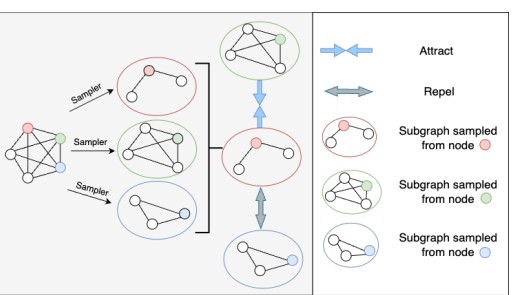

**Figure 5: Modeling subgraph-level correlations.**

adjacency matrix $A_i$ and feature matrix $X_i$ are as follows:

$$A_i = A_{idx,idx}, X_i = X_{idx,:},$$

where $.idx$ is an indexing operation. $A_{idx,idx}$ is the rowwise and col-wise indexed adjacency matrix corresponding to the induced subgraph. $X_{idx,:}$ is the row-wise indexed feature matrix.

**Encoding subgraph.** Given the context subgraph $G_i = (A_i, X_i)$ of a central node $i$, the encoder $\mathcal{E} : \mathbb{R}^{N \times N} \times \mathbb{R}^{N \times F} \to \mathbb{R}^{N \times F}$ encodes it to obtain the latent representations matrix $H_i$ denoted as

$$H_i = \mathcal{E}(A_i, X_i)$$

The subgraph-level representation $s_i$ is summarized by a readout function, $\mathcal{R} : \mathbb{R}^{N \times F} \to \mathbb{R}^F$:

$$s_i = \mathcal{R}(H_i)$$

.

So far, the representations of subgraphs have been produced. As shown in Figure 5, to model the correlations in subgraph level, we treat two subgraphs $s_i$ and $\hat{s}_i$ that sampled from the node $h_i$ and its most important neighbor node $\hat{h}_i$ respectively as positive pairs while the rest of subgraphs $\widetilde{s}$ are negative pairs. We minimize the following Subgraph-level Contrastive (SC) loss:

$$\mathcal{L}_{SC} = \frac{1}{M}\Sigma_{m=1}^M \mathcal{L}_{HFC}(s_m, \hat{s}_m, \widetilde{s_m}) \qquad (5)$$

*4.2.2 Inter-hierarchy contrastive learning.*
Having modeled the intra-hierarchy correlations, we further consider modeling the intra-hierarchy correlations as different hierarchies are dependent and will influence each other.
**Modeling Token-Node Correlations.** To model the token-node correlation, our intuition is to train the language model to refine the understanding of the text by GNN produced embeddings. Therefore, the language model is pushed to learn fine-grained task-aware context information. Specifically, given a sequence $x_v = \{x_{v,1}, x_{v,2}, ..., x_{v,T}\}$, we consider $x_v$ and its corresponding node representation $h_v$ as a positive pair. On the other hand, for a set of node representations, we employ a function, $\mathcal{P}$, to corrupt them to generate negative samples, denoted as

$$\{\widetilde{h_1}, \widetilde{h_2}, ..., \widetilde{h_M}\} = \mathcal{P}\{h_1, h_2, ..., h_M\},$$

where $M$ is the size of the representation set. $\mathcal{P}$ is the random shuffle function in our experiment. This corruption strategy determines the differentiation of tokens with different context nodes, which is

crucial for some downstream tasks, such as node classification. We develop the following Token-Node Contrastive (TNC) loss:

$$\mathcal{L}_{TNC} = \frac{1}{M}\Sigma_{m=1}^{M}\mathcal{L}_{HFC}(x_{m,v}, h_{m,v}, \mathcal{P}\{h_1, h_2, ..., h_M\}) \quad (6)$$

**Modeling Node-Subgraph Correlations.** Intuitively, nodes are dependent on their regional neighborhoods and different nodes have different context subgraphs. Therefore, we consider the strong correlation between central nodes and their context subgraphs to design a self-supervision pretext task to contrast the real context subgraph with a fake one. Specifically, for the node representation, $h_v$, that captures the regional information in the context subgraph, we regard the context subgraph representation $s_v$ as the positive sample. Similar to the calculation of $\mathcal{L}_{TNC}$, we employ the random shuffle function $\mathcal{P}$ to corrupt other subgraph representations to generate negative samples, denoted as

$$\{\widetilde{s_1}, \widetilde{s_2}, ..., \widetilde{s_M}\} = \mathcal{P}\{s_1, s_2, ..., s_M\}$$

We minimize the following Node-Subgraph Contrastive (NSC) loss:

$$\mathcal{L}_{NSC} = \frac{1}{M}\Sigma_{m=1}^{M}\mathcal{L}_{HFC}(h_{m,v}, s_{m,v}, \mathcal{P}\{s_1, s_2, ..., s_M\}) \quad (7)$$

**Overall Objective Loss.** Our overall objective function is a weighted combination of the five terms above:

$$\mathcal{L}_{HASH-CODE} = \lambda_{TC}\mathcal{L}_{TC} + \lambda_{NC}\mathcal{L}_{NC} + \lambda_{SC}\mathcal{L}_{SC} \\ + \lambda_{TNC}\mathcal{L}_{TNC} + \lambda_{NSC}\mathcal{L}_{NSC}, \quad (8)$$

where $\lambda_{TC}$, $\lambda_{NC}$, $\lambda_{SC}$, $\lambda_{TNC}$ and $\lambda_{NSC}$ are hyper-parameters that balance the contribution of each term. We summarize the workflow of our proposed HASH-CODE in Appendix B.3.

# 5 EXPERIMENTS

## 5.1 Experimental Setup

In this section, we have conducted extensive experiments, and analyzed the performance of the proposed HASH-CODE method by addressing the following key research questions as follows:

- **RQ1:** How does our method perform compared with baseline methods?
- **RQ2:** How does each component of our method contribute to the performance?
- **RQ3:** How about the efficiency of our proposed model compared with other baselines?
- **RQ4:** How does our method perform when facing the issue of data sparsity?
- **RQ5:** How do different hyper-parameters affect our method?

*5.1.1 Datasets.* We conduct experiments on six datasets (*i.e.,* DBLP[1], Wikidata5M[2] [51], Beauty, Sports and Toys from Amazon dataset[3] [35] and Product Graph) from three different domains (i.e., academic papers, social media posts, and e-commerce). We leverage three common metrics to measure the prediction accuracy: Precision@1 (P@1), NDCG, and MRR. Detailed information about the datasets can be found in Appendix B.1. The statistics of the six datasets are summarized in Table 1.

---

[1]https://originalstatic.aminer.cn/misc/dblp.v12.7z
[2]https://deepgraphlearning.github.io/project/wikidata5m
[3]http://snap.stanford.edu/data/amazon/

**Table 1: Statistics of datasets after preprocessing.**

| Dataset | Product | Beauty | Sports | Toys | DBLP | Wiki |
|---------|---------|--------|--------|------|------|------|
| #Users | 13,647,591 | 22,363 | 25,598 | 19,412 | N/A | N/A |
| #Items | 5,643,688 | 12,101 | 18,357 | 11,924 | 4,894,081 | 4,818,679 |
| #N | 4.71 | 8.91 | 8.28 | 8.60 | 9.31 | 8.86 |
| #Train | 22,146,934 | 188,451 | 281,332 | 159,111 | 3,009,506 | 7,145,834 |
| #Valid | 30,000 | 3,770 | 5,627 | 3,182 | 60,000 | 66,167 |
| #Test | 306,742 | 6,280 | 9,377 | 5,304 | 100,000 | 100,000 |

*5.1.2 Baselines.* We compare HASH-CODE with three types of baselines: (1) GNN-cascaded transformers, which includes BERT+MaxSAGE [13], BERT+MeanSAGE [13], BERT+GAT [44], TextGNN [71], and AdsGNN [29]. (2) GNN-nested transformers, which includes GraphFormers [60], and Heterformer [22]. (3) To verify the importance of both text and network information in TAGs, we also include Vanilla GraphSAGE [13], Vanilla GAT [44], Vanilla BERT [10] and Twin-Bert [34] in comparison. Detailed information about the baselines can be found in Appendix B.2.

*5.1.3 Reproducibility.* For all compared models, we adopt the 12-layer BERT-base-uncased [10] in the huggingface as the backbone PLM for a fair comparison. The models are trained for at most 100 epochs on all datasets. We use an early stopping strategy on P@1 with a patience of 2 epochs. The size of minimal training batch is 64, learning rate is set to $1e-5$. We pad the sequence length to 32 for Product, DBLP and Amazon datasets, 64 for Wiki, depending on different text length of each dataset. Adam optimizer [26] is employed to minimize the training loss. Other parameters are tuned on the validation dataset and we save the checkpoint with the best validation performance as the final model. Parameters in baselines are carefully tuned on the validation set to select the most desirable parameter setting.

## 5.2 Overall Comparison (RQ1)

Following previous studies on network representation learning, we consider two fundamental tasks: link prediction, node classification. To save space, we will mainly present the results on link prediction here and save the node classification part to Appendix C:

**Settings.** The link prediction experiments are evaluated in terms of link prediction accuracy, i.e., to predict whether a query node and key node are connected given the textual features of themselves and their neighbours. For Product, DBLP and Wiki datasets, in each testing instance, one query is provided with 300 keys: 1 positive plus 299 randomly sampled negative cases.

**Results.** The overall evaluation results are reported in Table 2. We have the following observations:

For four vanilla textual/graph baselines, the performance order is consistent across all datasets, *i.e.,* Bert > Twin-Bert > GAT ≈ GraphSAGE. GNN models obtain the worst performance, as they can only model the node proximity that preserved by the global structural information, but fail to encode the textual information that presents rich semantics to characterize the property of each node. This demonstrates the importance of leveraging the local textual information of individual nodes. As for the vanilla textual

**Table 2: Experiment results of link prediction. The results of the best performing baseline are underlined. The numbers in bold indicate statistically significant improvement (p < .01) by the pairwise t-test comparisons over the other baselines.**

| Datasets | Metric | MeanSAGE | GAT | Bert | Twin-Bert | Bert+MeanSAGE | Bert+MaxSAGE | Bert+GAT | TextGNN | AdsGNN | GraphFormers | Heterformer | HASH-CODE | Improv. |
|---|---|---|---|---|---|---|---|---|---|---|---|---|---|---|
| Product | P@1 | 0.6071 | 0.6049 | 0.6563 | 0.6492 | 0.7240 | 0.7250 | 0.7270 | 0.7431 | 0.7623 | 0.7786 | 0.7820 | **0.7967**$^*$ | 1.88% |
| | NDCG | 0.7384 | 0.7401 | 0.7911 | 0.7907 | 0.8337 | 0.8371 | 0.8378 | 0.8494 | 0.8605 | 0.8793 | 0.8861 | **0.9039**$^*$ | 2.01% |
| | MRR | 0.6619 | 0.6627 | 0.7344 | 0.7285 | 0.7871 | 0.7832 | 0.7880 | 0.8107 | 0.8361 | 0.8430 | 0.8492 | **0.8706**$^*$ | 2.52% |
| Beauty | P@1 | 0.1376 | 0.1367 | 0.1528 | 0.1492 | 0.1593 | 0.1586 | 0.1544 | 0.1625 | 0.1669 | 0.1774 | 0.1739 | **0.1862**$^*$ | 4.96% |
| | NDCG | 0.2417 | 0.2469 | 0.2702 | 0.2683 | 0.2741 | 0.2756 | 0.2726 | 0.2863 | 0.2891 | 0.2919 | 0.2911 | **0.3061**$^*$ | 4.86% |
| | MRR | 0.2558 | 0.2549 | 0.2680 | 0.2638 | 0.2712 | 0.2759 | 0.2720 | 0.2802 | 0.2821 | 0.2893 | 0.2841 | **0.3057**$^*$ | 5.67% |
| Sports | P@1 | 0.1102 | 0.1088 | 0.1275 | 0.1237 | 0.1330 | 0.1311 | 0.1302 | 0.1421 | 0.1466 | 0.1548 | 0.1534 | **0.1623**$^*$ | 4.84% |
| | NDCG | 0.2091 | 0.2116 | 0.2375 | 0.2297 | 0.2432 | 0.2478 | 0.2419 | 0.2537 | 0.2582 | 0.2674 | 0.2692 | **0.2775**$^*$ | 3.08% |
| | MRR | 0.2171 | 0.2168 | 0.2319 | 0.2296 | 0.2434 | 0.2471 | 0.2397 | 0.2612 | 0.2653 | 0.2679 | 0.2640 | **0.2754**$^*$ | 2.80% |
| Toys | P@1 | 0.1342 | 0.0.1330 | 0.1498 | 0.1427 | 0.1520 | 0.1536 | 0.1514 | 0.1658 | 0.1674 | 0.1703 | 0.1685 | **0.1767**$^*$ | 3.76% |
| | NDCG | 0.2015 | 0.2028 | 0.2249 | 0.2206 | 0.2451 | 0.2486 | 0.2413 | 0.2692 | 0.2734 | 0.2859 | 0.2823 | **0.2946**$^*$ | 3.04% |
| | MRR | 0.2173 | 0.2149 | 0.2311 | 0.2276 | 0.2509 | 0.2527 | 0.2476 | 0.2648 | 0.2715 | 0.2803 | 0.2778 | **0.2919**$^*$ | 4.14% |
| DBLP | P@1 | 0.4963 | 0.4931 | 0.5673 | 0.5590 | 0.6533 | 0.6596 | 0.6634 | 0.6913 | 0.7102 | 0.7267 | 0.7288 | **0.7446**$^*$ | 2.17% |
| | NDCG | 0.6997 | 0.6981 | 0.7484 | 0.7417 | 0.8004 | 0.8059 | 0.8086 | 0.8331 | 0.8507 | 0.8565 | 0.8576 | **0.8823**$^*$ | 2.88% |
| | MRR | 0.6314 | 0.6309 | 0.6777 | 0.6643 | 0.7266 | 0.7067 | 0.7300 | 0.7792 | 0.7805 | 0.8133 | 0.8148 | **0.8428**$^*$ | 3.44% |
| Wiki | P@1 | 0.2850 | 0.2862 | 0.3066 | 0.3015 | 0.3306 | 0.3264 | 0.3412 | 0.3693 | 0.3820 | 0.3952 | 0.3947 | **0.4104**$^*$ | 3.85% |
| | NDCG | 0.5389 | 0.5357 | 0.5699 | 0.5613 | 0.5730 | 0.5737 | 0.6071 | 0.6098 | 0.6155 | 0.6230 | 0.6233 | **0.6402**$^*$ | 2.71% |
| | MRR | 0.4411 | 0.4436 | 0.4712 | 0.4602 | 0.4980 | 0.4970 | 0.5022 | 0.5097 | 0.5134 | 0.5220 | 0.5216 | **0.5356**$^*$ | 2.61% |

baselines, the one-tower textual model (BERT) outperforms the two-tower model (Twin-BERT) as it can incorporate the information from both sides, while two-tower models can only exploit the data from a single side. However, one-tower structure has to compute the similarity between a search query and each ad one-by-one, which is not suitable for low-latency online scenario. In general, vanilla textual/graph models perform worse than GNN-cascaded transformers, which demonstrates the importance of encoding both text and network signals in text-attributed graphs.

As for GNN-cascaded transformers, Bert+GAT performs better than Bert+MeanSAGE and Bert+MaxSAGE on Product, DBLP and Wiki datasets, because the multi-head self-attention mechanism has a stronger capacity to model attributes. However, the performance of GAT is worse than that of MeanSAGE on Beauty, Sports and Toys datasets. A potential reason is that the multi-head self-attention may incorporate more noise from the attributes since they are keywords extracted from the reviews on Amazon Reviews. In general, GNN-cascaded transformers perform worse than co-training-based methods, which may be due to the node textual features are pre-existed and fixed in the training phase, leading to the limited expression capacity. AdsGNN consistently outperforms TextGNN on all datasets. This is because compared with TextGNN, the node-level aggregation model AdsGNN can capture the different roles of queries and keys, demonstrating that the tightly-coupled structure is more powerful than the loosely-coupled framework in deeply fusing the graph and textual information.

For GNN-nested transformers, Heterformer yields a larger performance improvement over Graphformers when network is more dense (i.e., Product and DBLP vs. Amazon datasets). By comparing our approach with all the baselines, it is clear to see that our

HASH-CODE performs consistently better than them with notable advantages on six datasets. Particularly, it achieves over 2%~4% relative improvements over the most competitive baselines (underlined) on each of the experimental datasets. Different from these baselines, we adopt the contrastive learning to enhance the representations of the attribute, and nodes for the representation learning task, which incorporates five pre-training objectives to model multiple data correlations by our proposed HFC-aware contrastive objectives. This result also shows that the contrastive learning approach is effective to improve the performance of the co-training architecture for representation learning.

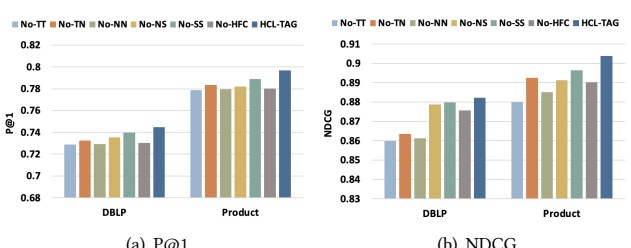

| (a) P@1 | (b) NDCG |

**Figure 6: Ablation studies of different components on DBLP and Products datasets.**

## 5.3 Ablation Study (RQ2)

Our proposed HASH-CODE designs five pre-training objectives based on the HFC-aware contrastive objective. In this section, we conduct the ablation study on Product and DBLP datasets to analyze

**Table 3: Time and memory costs per mini-batch for GraphFormers and HASH-CODE, with neighbour size increased from 3 to 200. HASH-CODE achieve similar efficiency and scalability as GraphFormers.**

| #N | 3 | 5 | 10 | 20 | 50 | 100 | 200 |
|---|---|---|---|---|---|---|---|
| Time: GraphFormers | 63.95ms | 97.19ms | 170.16ms | 306.12ms | 714.32ms | 1411.09ms | 2801.67ms |
| Time: HASH-CODE | 67.68ms | 105.35ms | 180.03ms | 324.11ms | 754.97ms | 1573.29ms | 2962.86ms |
| Mem: GraphFormers | 1.33GiB | 1.39GiB | 1.55GiB | 1.83GiB | 2.70GiB | 4.28GiB | 7.33GiB |
| Mem: HASH-CODE | 1.33GiB | 1.39GiB | 1.55GiB | 1.84GiB | 2.72GiB | 4.43GiB | 7.72GiB |

the contribution of each objective. We evaluate the performance of several HASH-CODE variants: (a) No-TT removes the $\mathcal{L}_{TC}$; (b) No-TN removes the $\mathcal{L}_{TNC}$; (c) No-NN removes the $\mathcal{L}_{NC}$; (d) No-NS removes the $\mathcal{L}_{NSC}$; (e) No-SS removes the $\mathcal{L}_{SC}$; (f) No-HFC replaces the HFC-aware loss with spectral contrastive loss. The results from GraphFormers are also provided for comparison. P@1 and NDCG@10 are adopted for this evaluation.

From Figure 6, we can observe that removing any contrastive learning objective would lead to the performance decrease, indicating all the objectives are useful to capture the correlations in varying levels of granularity in TAGs. Besides, the importance of these objectives is varying on different datasets. Overall, $\mathcal{L}_{TC}$ is more important than others. Removing it yields a larger drop of performance on all datasets, indicating that natural language understanding is more important on these datasets. In addition, No-HFC performs worse than the other variants, indicating the importance of learning more discriminative embeddings with HFC.

It is clearly seen that all model variants are better than Graph-Formers, which is trained only with link predication loss.

### 5.4 Efficiency Analysis (RQ3)

We compare the time efficiency between HASH-CODE, and GNN-nested Transformers (GraphFormers). The evaluation is conducted utilizing an Nvidia 3090 GPU. We follow the same setting with [60], where each mini-batch contains 32 encoding instances; each instance contains one center and #N neighbour nodes; the token length of each node is 16. We report the average time and memory (GPU RAM) costs per mini-batch in Table 3.

We find that the time and memory costs associated with these methods exhibit a linear escalation in tandem with the augmentation of neighboring elements. Meanwhile, the overall time and memory costs of HASH-CODE exhibit a remarkable proximity to GraphFormers, especially when the number of neighbor nodes is small. In light of the above observations, it is reasonable to deduce that HASH-CODE exhibits superior accuracy while concurrently maintaining comparable levels of efficiency and scalability when juxtaposed with GNN-nested transformers.

### 5.5 In-depth Analysis (RQ4 & 5)

We continue to investigate several properties of the models in the next couple sections. To save space, we will mainly present the results here and save the details to the appendix:
- In Appendix D.1, we simulate the data sparsity scenarios by using different proportions of the full dataset. We find that HASH-CODE is consistently better than baselines in all cases, especially

in an extreme sparsity level (20%). This observation implies that HASH-CODE is able to make better use of the data with the contrastive learning method, which alleviates the influence of data sparsity problem for representation learning to some extent.
- In Appendix D.2, we investigate the influence of the number of training epochs on our performance. The results show that our model benefits mostly from the first 20 training epochs. And after that, the performance improves slightly. Based on this observation, we can conclude that the correlations among different views on TAGs can be well-captured by our contrastive learning approach through training within a small number of epochs. So that the enhanced data representations can improve the performance of the downstream tasks.
- In Appendix D.3, we analyze the impact of neighbourhood size with a fraction of neighbour nodes randomly sampled for each center node. We can observe that with the increasing number of neighbour nodes, both HASH-CODE and Graphformers achieve higher prediction accuracies. However, the marginal gain is varnishing, as the relative improvement becomes smaller when more neighbours are included. In all the testing cases, HASH-CODE maintains consistent advantages over Graph-Formers, which demonstrates the effectiveness of our proposed method.
- In Appendix D.4, we visualize the input node embeddings for different target classes by t-SNE [43] to intuitively study the impact of our HFC-loss. We find that our $\mathcal{L}_{HFC}$ helps the model learn more discriminative node embeddings compared with $\mathcal{L}_{Spectral}$.

## 6 CONCLUSION

In this paper, we introduce the problem of node representation learning on TAGs and propose HASH-CODE, a hierarchical contrastive learning architecture to address the problem. Different from previous "cascaded architectures", HASH-CODE utilizes five self-supervised optimization objectives to facilitate thorough mutual enhancement between network and text signals in different granularities. We also propose a HFC-aware spectral contrastive loss to learn more discriminative node embeddings. Experimental results on various graph mining tasks, including link prediction and node classification demonstrate the superiority of HASH-CODE. Moreover, the proposed framework can serve as a building block with different task-specific inductive biases. It would be interesting to see its future applications on real-world TAGs such as recommendation, abuse detection and tweet-based network analysis.

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

## A  THEORETICAL ANALYSIS OF HFC

### A.1  Background: Spectral Clustering

Given a graph $\mathcal{G} = (\mathcal{V}, \mathcal{E})$, with adjacency matrix $A$, the Laplacian matrix of the graph is defined as $L = D - A$, where $D = diag(d_1, ..., d_N)$ is the diagonal degree matrix ($d_i = \Sigma_j A_{i,j}$). Then the symmetric normalized Laplacian matrix is defined as $L_{sym} = D^{-\frac{1}{2}} L D^{-\frac{1}{2}}$. As $L_{sym}$ is real symmetric and positive semidefinite, therefore it can be diagonalized as $L = U \Lambda U^T$ [9]. Here $U \in \mathbb{R}^{N \times N} = [u_1, ..., u_N]$, where $u_i \in \mathbb{R}^N$ denotes the $i$-th eigenvector of $L_{sym}$ and $\Lambda = diag(\lambda_1, ..., \lambda_N)$ is the corresponding eigenvalue matrix. To partition the graph, spectral clustering [16, 46] computes the first K eigenvectors and creates a feature vector $f_{K,v} \in \mathbb{R}^K$ for each node $v : \forall k \in [1, K], f_{K,v}(k) = u_k(v)$, which is in turn used to obtain K clusters by K-means or hierarchical clustering, etc.

An analogy between signals on graphs and usual signals [40] suggests to interpret the spectrum of $L_{sym}$ as a Fourier domain for graphs, hence defining filters on graphs as diagonal operators after change of basis with $U^{-1}$. It turns out that the features $f_{K,v}$ can be obtained by ideal low-pass filtering of the Delta function $\delta_a$ (localized at node a). Indeed, let $l_K$ be the step function where $l_K(\lambda) = 1$ if $\lambda < \lambda_K$ and 0 otherwise. We define $L_K$ the diagonal matrix for which $L_K(i, i) = l_K(\lambda_i)$. Then we have: $f_{K,v} = L_K U^{-1} \delta_v \in \mathbb{R}^K$, where we fill the last $N - K$ values with 0's. Therefore, spectral clustering is equivalent to clustering using low-pass filtering of the local descriptors $\delta_v$ of each node $v$ of the graph $\mathcal{G}$.

### A.2  Spectral Contrastive Loss Revisited

To introduce spectral contrastive loss [14], we give the definition of population view graph [14] first.

**Population View Graph.** A population view graph is defined as $\mathcal{G} = (\mathcal{X}, \mathcal{W})$, where the set of nodes comprises all augmented views $\mathcal{X}$ of the population distribution, with $w_{xx'} \in \mathcal{W}$ the edge weights of the edges connecting nodes $x, x'$ that correspond to different views of the same input datapoint. The core assumption made is that this graph cannot be split into a large number of disconnected subgraphs. This set-up aligns well with the intuition that in order to generalize, the contrastive notion of "similarity" must extent beyond the purely single-instance-level, and must somehow connect distinct inputs points.

**Spectral Contrastive Loss.** Using the concept of population view graph, spectral contrastive loss is defined as:

$$\mathcal{L}(x, x^+, x^-, f_\theta) = -2 \cdot \mathbb{E}_{x,x^+}[f_\theta(x)^T f_\theta(x^+)] \\ + \mathbb{E}_{x,x^-}[(f_\theta(x)^T f_\theta(x^-))^2], \tag{9}$$

where $(x, x^+)$ is a pair of views of the same datapoint, $(x, x^-)$ is a pair of independently random views, and $f_\theta$ is a parameterized function from the data to $\mathbb{R}^k$. Minimizing spectral contrastive loss is equivalent to spectral clustering on the population view graph, where the top smallest eigenvectors of the Laplacian matrix are preserved as the columns of the final embedding matrix $F$.

### A.3  HFC-aware Spectral Contrastive Loss

As discussed in Appendix A.2, the spectral contrastive loss only learns the low-frequency component (LFC) of the graph from a spectral perspective, where the effects of high-frequency components

(HFC) are much more attenuated. Recent studies have indicated that the LFC does not necessarily contain the most crucial information; while HFC may also encode useful information that is beneficial for the performance [2, 7]. In this regard, merely using the spectral contrastive loss cannot adequately capture the varying significance of different frequency components, thus constraining the expressiveness of learned representations and producing suboptimal learning performance. How to incorporate the HFC to learn a more discriminative embedding still requires explorations.

In image signal processing, the Laplacian kernel is widely used to capture high-frequency edge information for various tasks such as image sharpening and blurring [18]. As its counterpart in Graph Signal Processing (GSP) [40], we can multiply the graph Laplacian matrix $L$ with the input graph signal $x \in \mathbb{R}^N$, (i.e., $h = Lx$) to characterize its high-frequency components – the frequencies that carry sharply varying signal information across edges of graph. On the contrary, when highlighting the LFC, we would subtract the term $Lx$ which emphasizes more on HFC from the input signal $x$, i.e., $z = x - Lx$.

It should be noted that the above operation corresponds to a fixed low-pass filter in the spectral domain, where higher weights are specified for LFC. However, in practice, LFC may not always be useful, and HFC can also provide complementary insights for learning [2, 7], especially when the label information is not smooth across edges. Additionally, the HFC of the input graph signal would be unavoidably too much weakened compared with the lower ones with fixed filters, leading to the well-known over-smoothing problem [31]. As discussed in Appendix A.1, spectral clustering is equivalent to clustering using a low-pass filter on each node of the graph. Henceforth, the feature vectors learned by the spectral contrastive loss is LFC of the population view graph. In this regard, the fixed low-pass filters largely limit the fitting capability of contrastive learning and its variants for learning discriminative node representations. As a consequence, it is vital to capture the varying importance of frequencies in the filter to preserve more useful information and alleviate over-smoothing issues.

As an alternative of the traditional low-pass filter, a simple and elegant solution to introduce HFC is to assign a single parameter to control the rate of high-frequency substraction.

$$z = x - \alpha Lx = (I - \alpha L)x,$$

where $I$ is the identity matrix. We thus obtain the kernel $I - \alpha L$ that contains HFC.

Following [14], we consider the following matrix factorization based objective for eigenvectors:

$$\min_{F \in \mathbb{R}^{N \times K}} \mathcal{L}_{mf}(F) = ||(I - \alpha L) - FF^T||_F^2$$
$$= ((1-\alpha)I + \Sigma_{i,j}(\frac{\alpha w_{x_i, x_j}}{\sqrt{w_{x_i}} \sqrt{w_{x_j}}} - f_\theta(x_i)^T f_\theta(x_j)))^2, \tag{10}$$

where $w_x = \Sigma_{x' \in \mathcal{X}} w_{xx'}$ is the total weights associated to view $x$. By the classical low-rank approximation theory (Eckart-Young-Mirsky theorem [11]), minimizer $F$ possesses eigenvectors of HFC-aware kernel $I - \alpha L$ as columns and thus contains both the LFC and HFC of the population view graph.

LEMMA 1. *(HFC-aware spectral contrastive loss.) Denote $p_x$ is the x-th row of F. Let $p_x = w_x^{1/2} f_\theta(x)$. Then, the loss function $\mathcal{L}_{mf}(F)$ is equivalent to the following loss function for $f_\theta$, called HFC-aware spectral contrastive loss, up to an additive constant:*

$$\mathcal{L}_{mf}(F) = \mathcal{L}_{HFC}(f_\theta) + const,$$

*where*

$$\mathcal{L}_{HFC}(f_\theta) = -2\alpha \mathbb{E}_{x,x^+}[f_\theta(x)^T f_\theta(x^+)] \\ + \mathbb{E}_{x,x^-}[(f_\theta(x)^T f_\theta(x^-))^2] \quad (11)$$

PROOF. We expand $\mathcal{L}_{mf}(F)$ and obtain

$$\mathcal{L}_{mf}(F) = ((1-\alpha)I + \Sigma_{i,j}(\frac{\alpha w_{x_i,x_j}}{\sqrt{w_{x_i}}\sqrt{w_{x_j}}} - f_\theta(x_i)^T f_\theta(x_j)))^2$$

$$= const - 2\Sigma_{i,j}[(1-\alpha)I + \frac{\alpha w_{x_i,x_j}}{\sqrt{w_{x_i}}\sqrt{w_{x_j}}}]f_\theta(x_i)^T f_\theta(x_j)$$

$$+ \Sigma_{i,j}(f_\theta(x_i)^T f_\theta(x_j))^2$$

$$= \begin{cases} const - 2\Sigma_{i,j}1 - \alpha + \frac{\alpha w_{x_i,x_j}}{\sqrt{w_{x_i}}\sqrt{w_{x_j}}} f_\theta(x_i)^T f_\theta(x_j) \\ + \Sigma_{i,j}(f_\theta(x_i)^T f_\theta(x_j))^2, i = j \\ const - 2\Sigma_{i,j}\frac{\alpha w_{x_i,x_j}}{\sqrt{w_{x_i}}\sqrt{w_{x_j}}} f_\theta(x_i)^T f_\theta(x_j) \\ + \Sigma_{i,j}(f_\theta(x_i)^T f_\theta(x_j))^2, i \neq j \end{cases}$$

$$(12)$$

In our case two views $x_i$ and $x_j$ are not the same. We thus only focus on the $i \neq j$ case. Ignoring the scaling factor which doesn't affect linear probe error, we can hence rewrite the sum of last two terms of in Equation 12 as Equation 2. □

# B NOTES ON THE EXPERIMENTAL SETUP

## B.1 Details of Datasets

We conduct experiments on six datasets (*i.e.,* DBLP[4], Wikidata5M[5] [51], Beauty, Sports and Toys from Amazon dataset[6] [35] and Product Graph) from three different domains (i.e., academic papers, social media posts, and e-commerce):

**DBLP**: is a real-world academic citation graph dataset that contains the paper citation graph from DBLP up to 2020-04-09. Two papers are linked if one is cited by the other one. The paper's title is used as the textual feature.

**Wikidata5M** (Wiki): is a public entity graph dataset which contains the entity graph from Wikipedia. The first sentence in each entity's introduction is taken as its textual feature.

**Amazon Beauty, Sports and Toys**: are obtained from Amazon review datasets in [35], which contain product ratings and reviews in 29 categories on Amazon.com and rich textual metadata such as title, brand, description, etc. We use the version released in the year 2018. Specifically, we select three subcategories: "Beauty", "Sports and Outdoors", and "Toys and Games", and utilize the brands and the descriptions of the items as attributes. We treat all the user-item rating records as implicit feedback and sort them according to the timestamps to form sequences. Following the common settings [24], we filter out users and items with less than five interaction records. For each user, we use the last clicked item for testing,

---

[4]https://originalstatic.aminer.cn/misc/dblp.v12.7z
[5]https://deepgraphlearning.github.io/project/wikidata5m
[6]http://snap.stanford.edu/data/amazon/

the penultimate one for validation, and the remaining clicked items for training.

**Product Graph** (Product): is an even larger dataset of online products collected by a world-wide search engine. In this dataset, the users' web browsing behaviors are tracked for the targeted product webpages (e.g., Amazon webpages of Nike shoes). The user's continuously browsed webpages within a short period of time (e.g., 30 minutes) is called a "session". The products within a common session are connected in the graph (which is a common way of graph construction in e-commerce scenarios [48, 63]). Each product has its unique textual description, which specifies information like the product name, brand, and saler, etc.

The textual features of all the datasets are in English. We make use of uncased WordPiece [55] to tokenize the input text.

## B.2 Details of Baselines

To thoroughly examine the effectiveness of our proposed method and substantiate its validity, we contrast three types of competitive methods:

**First**, to verify the importance of both text and network information, we consider the vanilla textual/graph models that only exploit partial observed information (i.e., textual or structural) for node representation learning.

- Vanilla GraphSAGE [13]: This is a GNN method that employs the mean function to aggregate information from neighbors for center node embedding learning. The initial node feature vector is bag-of-words weighted by TF-IDF. The number of entries in each attribute vector corresponds to the vocabulary size of the respective dataset, where we retain the most representative 10000, 2000, and 5000 words for DBLP, Wiki, and Product, respectively, in accordance with the corpus size.
- Vanilla GAT [44]: Similar with the vanilla GraphSAGE, we employ the graph attention networks to aggregate the information from neighbors for center node embedding learning.
- Vanilla BERT [10]: This is a standard PLM pretrained on two tasks: next sentence prediction and mask token prediction. For each text-rich node, we use BERT to encode its text and extract the output of the [CLS] token as node representation.
- Twin-Bert [34]: This is a two-tower BERT-based structure model, which serves for the efficient retrieval.

**Second**, the GNN-cascaded transformers which combines the GNN and PLM in a "cascaded architectute" that learns the node representation with fixed textual embeddings.

- BERT+MaxSAGE [13]: We combine BERT with MaxSAGE (i.e., using the output text representation of BERT as the input node attribute vector of MaxSAGE). The BERT+MaxSAGE model is trained in an end-to-end manner. Other BERT+GNN baselines below have the same cascaded architecture.
- BERT+MeanSAGE [13]: MeanSAGE is a GNN method that applies the mean function during neighbor aggregation for center node representation learning.
- BERT+GAT [44]: GAT is a GNN method with attention-based neighbor importance calculation, and the weight of each neighbor during aggregation depends on its importance score.

- TextGNN [71]: This model incorporates the text and graph information with a node-level aggregator, in which the query encoders share the same parameters.
- AdsGNN [29]: This model also utilizes a node-level aggregator to aggregate the graph information at different levels, and introduces domain-specific pre-training and knowledge-distillation techniques to improve model performance.

**Third**, the state-of-the-art co-training-based methods that enables the joint encoding of text and node features for the node representation learning on TAGs.

- GraphFormers [60]: This is the state-of-the-art GNN-nested transformer model, which has graph-based propagation and aggregation in each transformer layer.
- Heterformer [22]: This model alternately stacks the graph-attention-based neighbor aggregation module and the transformer-based text and neighbor joint encoding module to facilitate thorough mutual enhancement between network and text signals.

## B.3 Summary of HASH-CODE's workflow

---

**Algorithm 1** HCL-TAG's Workflow

---

**Input:** The input graphs $G$ (consist of the center node $v$ and its neighbours).
**Output:** The embedding for the center node $h_v$.
**for** each text $g \in G$ **do**
    $H_g^1 \leftarrow \text{TRM}^0(H_g^0)$; // Get the initial token-level embeddings.
**end for**
**for** $l = 1, ..., L - 1$ **do**
    $Z_g^l \leftarrow \{z_g^l | g \in G\}$; // Gather node-level embeddings to GNN
    $\widetilde{Z}_g^l \leftarrow \text{GNN}(Z_g^l)$; // Graph aggregation in GNN
    **for** $i = 1, ..., 5$ **do**
        $\hat{Z}_g^l \leftarrow \text{Contrastive}_i(\widetilde{Z}_g^l, H_g^l)$; // Hierarchical contrastive learning for mutually reinforce the textual and graphic patterns
    **end for**
    **for** each text $g \in G$ **do**
        $\widetilde{H}_g^l \leftarrow \text{Concat}(\hat{z}_g^l, H_g^l)$; // Get contrastive graph-augmented token-level embeddings
        $H_g^{l+1} \leftarrow \text{TRM}^l(\widetilde{H}_g^l)$; // Text encoding in Transformer
    **end for**
**end for**
**return** $h_v \leftarrow \hat{z}_v^L$

---

## C NODE CLASSIFICATION

**Settings.** In node classification, we train a 2-layer MLP classifier to classify nodes based on the output node representation embeddings of each method. The experiment is conducted on DBLP. Following [22], we select the most frequent 30 classes in DBLP. Also, we study both transductive and inductive node classification to understand the capability of our model comprehensively. For transductive node classification, the model has seen the classified nodes during representation learning (using the link prediction objective), while for inductive node classification, the model needs to predict the label of nodes not seen before. We separate the whole dataset into train set, validation set, and test set in 7:1:2 in all cases and each experiment is repeated 5 times in this section with the average performance reported.

**Results.** Table 4 demonstrates the results of different methods in transductive and inductive node classification. We observe that: (a) our HASH-CODE outperforms all the baseline methods significantly on both tasks, showing that HASH-CODE can learn more effective node representations for these tasks; (b) GNN-nested transformers generally achieve better results than GNN-cascaded transformers, which demonstrates the necessity of introducing graphic patterns in modeling textual representations; (c) HASH-CODE generalizes quite well on unseen nodes as its performance on inductive node classification is quite close to that on transductive node classification. Moreover, HASH-CODE even achieves higher performance in inductive settings than the baselines do in transductive settings.

**Table 4: Experiment results of transductive and inductive node classification on DBLP dataset. (HASH-CODE marked in bold, the best baseline underlined). HASH-CODE outperforms all baselines, especially the ones based on GNN-nested transformers.**

| Model | Transductive | | Inductive | |
|---|---|---|---|---|
| | P@1 | NDCG | P@1 | NDCG |
| MeanSAGE | 0.5186 | 0.7231 | 0.5152 | 0.7197 |
| GAT | 0.5208 | 0.7196 | 0.5126 | 0.7146 |
| Bert | 0.5493 | 0.7506 | 0.5310 | 0.7485 |
| Twin-Bert | 0.5291 | 0.7440 | 0.5248 | 0.7431 |
| Bert+MeanSAGE | 0.6731 | 0.7637 | 0.6413 | 0.7494 |
| Bert+MaxSAGE | 0.6705 | 0.7752 | 0.6587 | 0.7599 |
| Bert+GAT | 0.6849 | 0.0.7801 | 0.6689 | 0.0.7619 |
| TextGNN | 0.6820 | 0.7753 | 0.6380 | 0.7716 |
| AdsGNN | 0.6882 | 0.7790 | 0.6624 | 0.7737 |
| GraphFormers | 0.6919 | 0.7929 | 0.6791 | 0.7993 |
| Heterformer | 0.6924 | 0.7957 | 0.6746 | 0.8079 |
| HASH-CODE | **0.7116** | **0.8198** | **0.6961** | **0.8170** |
| *Improv.* | 2.77% | 3.03% | 2.50% | 1.13% |

## D IN-DEPTH ANALYSIS

### D.1 Data Sparsity Analysis

Conventional representation learning methods require a considerable amount of training data, thus they are likely to suffer from the data sparsity issues in real-world applications. This problem can be alleviated by our method because the proposed contrastive learning approach can better utilize the data correlation from input. We simulate the data sparsity scenarios by using different proportions of the full dataset, i.e., 20%, 40%, 60%, 80%, and 100%.

Figure 7 shows the evaluation results on Product and Sports datasets. As we can see, the performance substantially drops when less training data is used. While, HASH-CODE is consistently better than baselines in all cases, especially in an extreme sparsity level (20%). This observation implies that HASH-CODE is able to make better use of the data with the contrastive learning method, which alleviates the influence of data sparsity problem for representation learning to some extent.

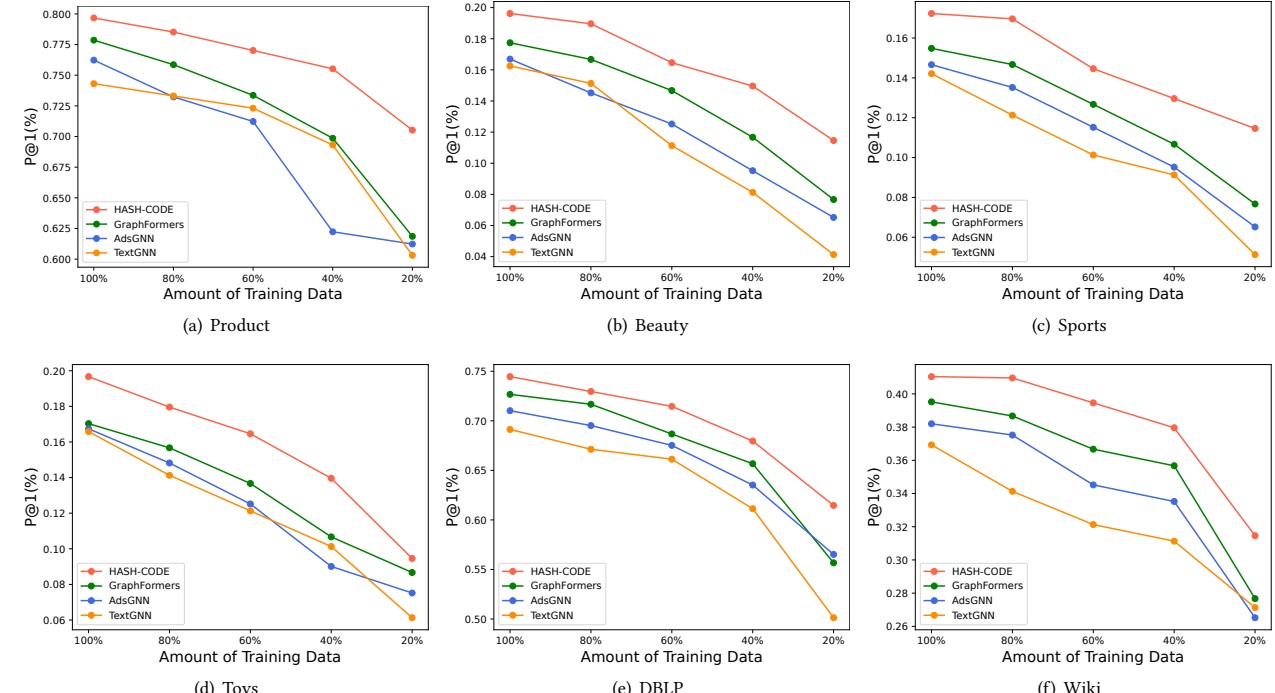

Figure 7: Performance (P@1) comparison w.r.t. different sparsity levels on DBLP and Product datasets. The performance substantially drops when less training data is used, while HASH-CODE is consistently better than baselines in all cases, especially in an extreme sparsity level (20%).

## D.2 Influence of Training Epochs Number

Our approach consists of co-training with GNNs and Transformers. During the training stage, our model can learn the enhanced representations of the attribute and node for the representation learning task. The number of training epochs will affect the performance of the downstream task. To investigate this, we train our model with a varying number of epochs and fine-tune it on the downstream task.

Figure 8 presents the results on Product and Sports datasets. We can see that our model benefits mostly from the first 20 training epochs. And after that, the performance improves slightly. Based on this observation, we can conclude that the correlations among different views (i.e., the graph topology and textual attributes) can be well-captured by our contrastive learning approach through training within a small number of epochs. So that the enhanced data representations can improve the performance of the downstream tasks.

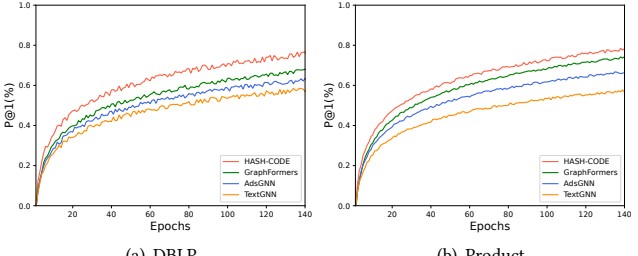

Figure 8: Performance (P@1) comparison w.r.t. different numbers of training epochs on DBLP and Product datasets. HASH-CODE benefits mostly from the first 20 training epochs, thus the correlations among different views can be well-captured by our approach through training within a small number of epochs.

## D.3 Influence of Neighbor Size

We analyze the impact of neighbourhood size with a fraction of neighbour nodes randomly sampled for each center node (using DBLP for illustration). The link prediction results are shown in Figure 9. We can observe that with the increasing number of neighbour nodes, both HASH-CODE and Graphformers achieve higher prediction accuracies. However, the marginal gain is varnishing,

as the relative improvement becomes smaller when more neighbours are included. In all the testing cases, HASH-CODE maintains consistent advantages over GraphFormers, which demonstrates the effectiveness of our proposed method.

## D.4 HFC-aware Embedding Visualization.

To intuitively study the impact of our HFC-loss, we visualize the input node embeddings for different target classes by t-SNE [43].

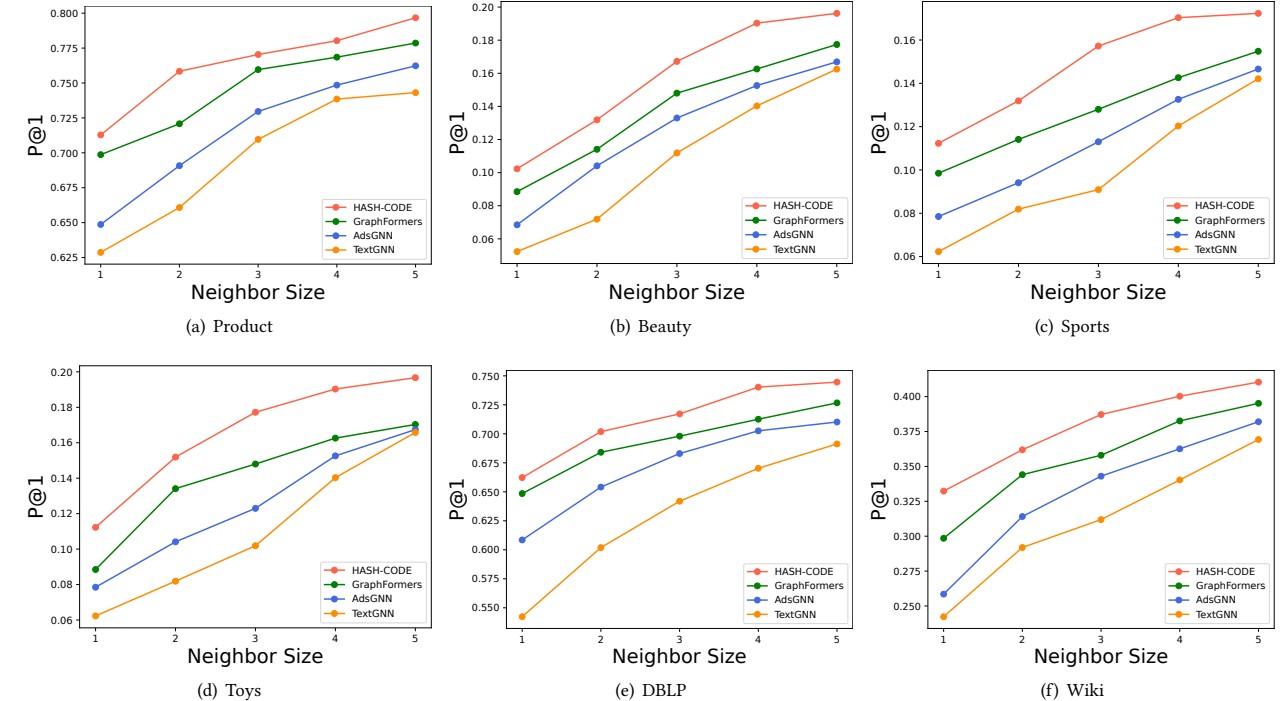

(a) Product        (b) Beauty        (c) Sports

(d) Toys        (e) DBLP        (f) Wiki

**Figure 9: Impact of neighbor size on DBLP dataset. Enlarging the number of neighbour nodes brings performance improvement to both models. HASH-CODE maintains consistent advantages over GraphFormers over all test cases.**

We conduct the visualization on DBLP with four different target classes, and each target class has more than 1000 node embeddings. Figure 10 shows that compared with HFC-aware loss, the spectral contrastive loss cannot effectively distinguish different types of sample nodes. Especially in the central part of Figure 10(a), sample points are almost completely overlapping. It is clear that the HFC-aware loss learns more discriminative node embeddings.

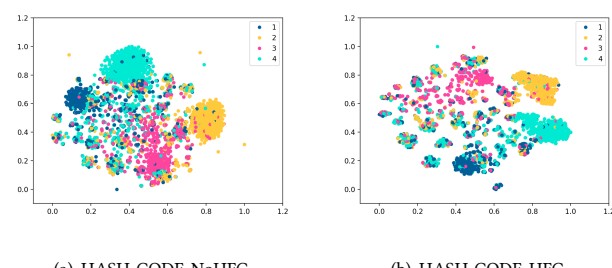

(a) HASH-CODE-NoHFC        (b) HASH-CODE-HFC

**Figure 10: Embedding visulization of input nodes belonging to different target classes. Points with the same color denote input nodes belonging to the same target class. HFC-aware loss learns more discriminative embeddings than spectral contrastive loss.**

