# OpenReview forum: "High-Frequency-aware Hierarchical Contrastive Selective Coding for Representation Learning on Text Attributed Graphs"
_ACM.org/TheWebConf/2024/Conference — TheWebConf24_

### Official Review · Reviewer_uVQq · 2023-11-10

**Novelty:** 5
**Technical Quality:** 5

**Review:**

--------------------------------------------
Updates after Authors' Response:

I have read the response from the authors, which have addressed most of my questions in the original review.

--------------------------------------------

The authors extend prior work for co-training GNNs and LMs within a unified learning framework for text-attributed graphs (TAGs).
The proposed approach utilized five self-supervised contrastive optimization objectives, targeting different forms or granularities of the input. In addition, each of the five contrastive losses if a modified version which is "high-frequency aware", which draws inspiration from spectral contrastive learning method.

Strengths:
- Novel unsupervised contrastive learning approach for joint-training of GNNs and LMs
- SoTA performance is demonstrated on six common datasets on both link prediction and node classification tasks
- The paper is generally well written

Weaknesses:
- Missing information on hyper-parameters optimization process and the ones eventually chosen
- Some claims are made without sufficient backing or context. E.g:
   - "these methods typically employ supervised training"
   -  "these PLM embeddings remain non-trainable during the GNN training phase"

Nit:
- "Remarkably, the primary distinction lies in the introduction of the parameter 𝛼 ."
	- Is not it the only difference between the two equations? if so, why describe it in this misleading way?
- "Results from GraphFormers are also provided for comparison."
	- are they? they are in the main table did not see it in figure 6

**Questions:**

- Are different entities (e.g., users and items) encoded using the same transformer?

- It is claimed that : "these PLM embeddings remain non-trainable during the GNN training phase".
Why this is the case? a longer discussion and some steps to enable such training are offered here: https://knowledge-nlp.github.io/aaai2023/papers/016-m5gnn-poster.pdf

- What were the values chosen for the hyper-parameters for each of the datasets? any learnings from that?

- Related question, in the ablation study, were the hyper-params optimized again? this could offset some of the degradation shown in ablations.

- The Heterformer paper is showing higher numbers in DBLP. What is the source of this difference?

**Reviewer Confidence:**

2: The reviewer is willing to defend the evaluation, but it is likely that the reviewer did not understand parts of the paper

**Scope:**

3: The work is somewhat relevant to the Web and to the track, and is of narrow interest to a sub-community

---

### Official Review · Reviewer_7N5Q · 2023-11-20

**Novelty:** 4
**Technical Quality:** 3

**Review:**

This paper studies the problem of representation learning on text attributed graphs using contrastive learning. Specifically, it discusses the benefits to employ high-frequency aware loss. To achieve this goal, the author devise three hierarchical losses and two intra-hierarchy losses. The results shows comparable efficiency with latest GNN co-training methods and better performance on link prediction task at six different datasets.

Personally, I won't champion this paper at the current state for several reasons:
1. Some technique details are not clearly written, such as model architectures and choices of hyper-parameters.
2. The connection between HFC and the motivation of five different posse are weak in my opinion.
3. The performance improvements are not that significant on only one tasks (link prediction) make the practical impact limited.

**Questions:**

1. What's the specific GNN architectures used in the paper and why it is used ?
2. Does GNNs and PLMs trained separately in contrastive learning, e.g. HASH-CODE is not cascaded and end-to-end trained ?
3. Can author briefly talk about what's the proposed model performance on node classification? Since node classification might be the another main tasks extensively studied on text attributed graphs.

**Reviewer Confidence:**

3: The reviewer is confident but not certain that the evaluation is correct

**Scope:**

3: The work is somewhat relevant to the Web and to the track, and is of narrow interest to a sub-community

---

### Official Review · Reviewer_fb8b · 2023-11-23

**Novelty:** 5
**Technical Quality:** 4

**Review:**

The paper proposes HASH-CODE, a novel approach for node representation learning on Text-Attributed Graphs. The authors seek to address the challenge of integrating GNNs and Pre-trained Language Models to capture both network and text signals effectively. The method employs a High-frequency Aware Spectral Hierarchical Contrastive Selective Coding method, utilizing five self-supervised optimization objectives to mutually enhance network and text signals in diverse granularities.

Pros:
- The paper introduces a contrastive learning approach that considers hierarchical intrinsic data correlations within TAGs, providing a better understanding of contextual information. The (HFC)-aware contrastive learning objective is very functional and novel since it ensures more distinctive embeddings by balancing low-frequency and high-frequency components.

Cons:
- The proposed model involves multiple optimization objectives, potentially adding complexity to implementation and training.
- No in-depth explanation on choice of GNNs or hyperparameters used.

**Questions:**

- How did you strike a balance between the complexity introduced by multiple optimization objectives and the practical implementation of your method?
- How does HASH-CODE compare to existing methods, especially those employing supervised training?

**Reviewer Confidence:**

4: The reviewer is certain that the evaluation is correct and very familiar with the relevant literature

**Scope:**

4: The work is relevant to the Web and to the track, and is of broad interest to the community

---

### Official Review · Reviewer_GpqN · 2023-11-24

**Novelty:** 5
**Technical Quality:** 5

**Review:**

**summary**

This paper proposes a novel GNN-nested Transformer to learn the text-attributed graph representation in an unsupervised manner. It designs five contrastive objectives to make node embedding more discriminative. The experiments on six benchmarks show that the model performs highlightly.

**pros**

1. The author notices that previous spectral contrastive learning has a core defect due to its low-pass property, and it proposes high-frequency-aware contrastive objective to avoid over-smoothing.

2. Sufficient experiments on different domain datasets show that the proposed method is effective.

3. From ablation study, it is clear that each contrastive objective is significant to the learning process.

**cons**

1. The motivation to utilize contrastive learning is not very clear. The problems of necessitating a substantial volume of labeled data and relying exclusively on limited optimization objectives are insufficient to relate closely to contrastive learning.

**Questions:**

Q1. How scalable is the proposed method? I think you need to add some experiments on different GNN backbones in you model.

**Reviewer Confidence:**

3: The reviewer is confident but not certain that the evaluation is correct

**Scope:**

3: The work is somewhat relevant to the Web and to the track, and is of narrow interest to a sub-community

---

### Official Review · Reviewer_qw98 · 2023-11-27

**Novelty:** 4
**Technical Quality:** 4

**Review:**

## Updates after Authors' Response
I have read the response from the authors, which have addressed most of my questions in the original review. I raise an additional question below for the authors' response. Overall, I deem it is fair to keep the scores in my original review.

---
## Summary

This work proposes to improve the representation learning on text-attributed graphs (TAGs) by (1) proposing five self-supervised contrastive loss terms that can be used in addition to the supervised optimization objective for training the GNN+transformer architecture, and (2) adopting a high-frequency component (HFC)-aware spectral contrastive learning loss function.

## Strengths

1. The question of effectively combining GNNs with transformer architectures to improve performance of representation learning on text-attributed graphs is important.
2. Comprehensive experiments are conducted with a wide-range of baselines that include GNN-nested transformers, upon which the proposed approach shows improvement (albeit not large).

## Weakness

1. **Without providing a workflow for systematic hyper-parameter tuning, the proposed approach involved many hyper-parameters** (specifically, the 5 $\lambda$-terms for balancing different losses and the $\alpha$ term in Eq. (2) that controls the ratio of high-frequency components) **that can be hard to tune in practice**. Moreover, the paper does not list the choices of these parameters used in the experiments, nor does it provide a sensitivity analysis.
2. **Some designs for the contrastive learning lacks justifications.** For example, for modeling the subgraph-level correlations, the authors first uses node-wise PPR to select the most important neighbor $u$ for each node $v$; the subgraph for node u is then deemed as a positive pair with the subgraph of $v$, while the remaining neighbors of $u$ is deemed deemed as negative pairs with $v$. However, the author did not justify (1) why use node-wise PPR to measure importance, and (2) why only the most important neighbor is being used to form the positive training pair here.

**Questions:**

Please address the comments in the weakness section.

**Reviewer Confidence:**

3: The reviewer is confident but not certain that the evaluation is correct

**Scope:**

4: The work is relevant to the Web and to the track, and is of broad interest to the community

---

### Decision · Program_Chairs · 2024-01-22

**Decision:**

Accept

**Comment:**

The paper addresses the coupling of graph neural networks (GNNs) and pretrained language models (PLMs) on text-attributed graphs (TAGs). The proposed HASH-CODE method integrates GNNs and PLMs through self-supervised optimization objectives, demonstrating efficacy in enhancing mutual understanding between network and text signals in diverse granularities, as validated by extensive experiments on real-world benchmarks.


 - Too many hyper-parameters without proper justification of those, limit the practical application of the method.
 - The motivation behind contrastive learning is not straightforward
 - Several relevant published works are not considered as comparisons.